# Body Condition Score Change throughout Lactation Utilizing an Automated BCS System: A Descriptive Study

**DOI:** 10.3390/ani12050601

**Published:** 2022-02-28

**Authors:** Carissa M. Truman, Magnus R. Campler, Joao H. C. Costa

**Affiliations:** 1Department of Animal and Food Sciences, University of Kentucky, Lexington, KY 40546, USA; carissa.truman@gmail.com (C.M.T.); campler.1@osu.edu (M.R.C.); 2Department of Veterinary Preventive Medicine, The Ohio State University, Columbus, OH 43210, USA

**Keywords:** precision dairy technology, prediction, 3D camera, automation

## Abstract

**Simple Summary:**

The aim of this study was to implement a commercially available automated body condition scoring (ABCS) camera system to collect data for developing a predictive equation of body condition dynamics throughout the lactation period. The body condition score can vary depending on many factors relative to a specific cow. Lactation number, DIM, disease status, and 305d-predicted-milk-yield (305PMY) were significant factors to create a multivariate prediction model for automatic body condition scores throughout lactation.

**Abstract:**

Body condition scoring (BCS) is a traditional visual technique often using a five-point scale to non-invasively assess fat reserves in cattle. However, recent studies have highlighted the potential in automating body condition scoring using imaging technology. Therefore, the objective was to implement a commercially available automated body condition scoring (ABCS) camera system to collect data for developing a predictive equation of body condition dynamics throughout the lactation period. Holstein cows (*n* = 2343, parity = 2.1 ± 1.1, calving BCS = 3.42 ± 0.24), up to 300 days in milk (DIM), were scored daily using two ABCS cameras mounted on sort-gates at the milk parlor exits. Scores were reported on a 1 to 5 scale in 0.1 increments. Lactation number, DIM, disease status, and 305d-predicted-milk-yield (305PMY) were used to create a multivariate prediction model for body condition scores throughout lactation. The equation derived from the model was: ABCS_ijk_ = 1.4838 − 0.00452 × DIM_i_ − 0.03851 × Lactation number_j_ + 0.5970 × Calving ABCS_k_ + 0.02998 × Disease Status(neg)_l_ − 1.52 × 10^−6^ × 305PMY_m_ + *e*_ijklm_. We identified factors which are significant for predicting the BCS curve during lactation. These could be used to monitor deviations or benchmark ABCS in lactating dairy cows. The advantage of BCS automation is that it may provide objective, frequent, and accurate BCS with a higher degree of sensitivity compared with more sporadic and subjective manual BCS. Applying ABCS technology in future studies on commercial dairies may assist in providing improved dairy management protocols based on more available BCS.

## 1. Introduction 

Body condition scoring (BCS) allows for an instantaneous appraisal of fat reserves of the cow. Fat reserves and changes in fat reserves over time are indicative of a cow’s energy balance [1]. Body condition scoring has been evaluated as a factor affecting many aspects on-farm and is used to make managerial changes. Traditionally, BCS is determined visually by staff or caretakers, leaving the accuracy of the scoring up to the training and experience of the individual scorer. However, despite similar training and experience, there may be differences in the degree of agreement between scorers’ (inter-rater reliability) BCS assessments for the same animal, as well as differences between the scorers’ own observations (intra-rater reliability) when revisiting the same animal in a short time frame. Thus, accurate scoring requires time and training to provide qualitative data with limited influence of subjectivity [1,2].

The time around parturition influences the BCS of dairy cattle. Commonly, cows around parturition reduce their fat reserves by 30% to 40% [3]. Negative energy balance, or when the cow’s nutritional demands exceed intake, is experienced by over 80% of dairy cows during each lactation [4,5]. Cows lose condition in the beginning of lactation, followed by a slow gain of condition as energy demands decrease throughout mid lactation [6]. It has been reported that cows, on average, reach a positive energy balance by 45 days in milk (DIM) and that 90% of cows are in positive energy balance by 63 DIM [7]. Cows tend to exert resources towards milk production until the next pregnancy, when the resumption of body reserves occur [8]. Thus, the greatest body condition loss occurs in the first 30 DIM, whereafter cows tend to maintain their condition until 90 DIM, when they start to regain body condition [9]. Although cows usually tend to lose body condition rapidly during the first 30 to 40 DIM, some studies have found that cows have their nadir BCS between 60 to 80 DIM and even up to 100 DIM [10,11]. Body condition can be affected by parity, DIM, and previous BCS [12]. Prior studies have shown that primiparous cows and second lactation cows have higher body condition scores compared to older cows and decreasing body condition scores with increased parity [13,14,15]. In terms of body condition loss, second lactation cows may lose significantly less body condition within 7 weeks post-calving compared to older cows, and multiparous cows may have greater losses in body condition from dry to nadir compared to primiparous cows [11,16,17]. Moreover, primiparous cows tend to lose less body condition compared to multiparous cows during the first months after parturition [18]. However, previous studies have reported that primiparious cows lose condition quicker and reach nadir sooner, while having a quicker condition recovery rate post-nadir compared to multiparous cows [12,17]. Finally, there are reports of no relationship between body condition loss and parity, adding ambiguity to the reliability of the currently reported BCS [19]. Cows that develop transition diseases have metabolic changes and decreased dry matter intake (DMI), which are both influential to body condition [20]. Cows that are diseased lose more body condition compared to healthy cows at 4 weeks post-calving [21]. A recent study showed that cows with ketosis had an extended period of body condition loss and had a higher body condition during the first 2 weeks in milk prior to or during the disease bout [14]. Although contrary findings regarding the relationship between diseased cows and BCS have been reported, most evidence supports an association between BCS in dairy cows and disease occurrence [18,22]. The opportunity to clarify previous contrasting findings on body condition fluctuations by increased BCS frequency and accuracy is important and should be pursued.

Selection for milk production has been strongly associated with increased body reserves mobilization [23,24,25]. Low BCS has been reported to be related to higher milk production [23]. However, a non-linear relationship between calving BCS and milk yield has been observed, where a calving BCS around 3.5 produced the highest 60-d yield [26]. These results are contrary to earlier findings suggesting that BCS is uncorrelated with milk yield [7,27]. The condition of cows at the start of lactation can impact their future progression and body condition curve. It has been observed that higher conditioned cows have a decreased DMI in the start of the lactation period, resulting in a greater loss of body condition as lactation progresses [6]. Moreover, cows with an underlying issue such as chronic or sub-acute rumen acidosis may be more likely to have a low BCS or experience delayed BCS recovery [28], and it has been observed that the body condition loss accelerates in over-conditioned cows compared to lower conditioned cows [22,29]. Additionally, over-conditioned cows at calving have been described to reach maximum DMI later in the lactation period compared to under-conditioned cows at calving [30]. It has been reported that both over- and under-conditioned cows pre-calving are likely to maintain their high and low body condition, respectively, throughout the first 5 months post-calving [31]. It has been assumed that a body condition change of 0.25 points or less would be undetectable using manual visual observation techniques after 3 weeks in lactation [7]. Researchers have modeled body condition across DIM using manual BCS data, but the limitation of time and labor reduce the frequency of data collected per animal [26,32,33]. The utilization of an automated body condition scoring (ABCS) system increases the frequency of BCS for individual cows, enabling ABCS to be more accurate compared to manually obtained scores. Thus, the chance for earlier detection of cows at risk is improved and may therefore facilitate management decisions. A growing number of studies have investigated different approaches for improving the objectivity and accuracy of BCS scoring. These approaches include still photos, thermal imaging, light-depth cameras, and 3D technology [34,35,36,37,38]. However, most novel technologies lack full automation and are not yet suitable for commercial farm implementation. The commercially available ABCS camera used in this study has previously been validated and field-tested, making it a reliable option for the data collection needed for the modelling of a predictive ABCS model [39]. The aim of this study was to determine the different trajectories of BCS derived from ABCS cameras by investigating a multivariate modelling approach including variables such as DIM, lactation number, calving ABCS, calving month, disease, and 305-d predicted milk yield (305PMY).

## 2. Materials and Methods 

This study took place on a commercial dairy farm in Indiana, USA, housing approximately 3200 dry and lactating Holstein cows. This is a retroactive trial based on data throughout one lactation cycle from a commercial farm, and no Institutional Animal Care and Use Committee of the University of Kentucky review was deemed necessary. From the general herd, 2343 Holstein cows were enrolled. Cows had a mean (±SD) parity of 2.1 ± 1.1, DIM of 186.1 ± 111.1, ABCS at calving of 3.42 ± 0.24, and 305PMY of 12,720 ± 2028 kg of milk). All cows were fed a total mixed ration adjusted for individual needs per the farm’s standard feeding operating protocols, which adjusted feed allocation based on desired BCS at various stages throughout lactation.

The farm had two automatically recording ABCS cameras (DeLaval Body Condition Scoring BCS, DeLaval International AB, Tumba, Sweden), with one mounted on top of each sort-gate at the two existing parlor exits. A radio frequency identification sensor (RFID) was attached to each sort-gate to allow for individual identification of cows exiting the parlor using their corresponding RFID ear tag. All lactating cows individually passed under the cameras once per day after their first morning milking but before feeding, thus allowing their ABCS to be obtained prior to any recent rumen fill. The camera system operated by filming a continuous 3D video (30 FPS, 32,000 projected IR reference points) from which a unique 3D image was created for each cow via an infrared projection pattern on the back of the cow. The projection patterns highlighted the peaks and valleys, created by muscle and body fat, across the cow’s back. The 3D image was then processed by the accompanying camera software, which used a proprietary algorithm, to generate an ABCS based on key physical characteristics (pins, tailhead ligaments, thurls, sacral ligaments, short ribs, and hooks) on the back of the cow [1]. The continuous recording and the camera software allowed for cows to be properly scored, regardless of whether they were standing still or in locomotion when passing under the camera, ensuring an ABCS for each passage. The scoring was reported in 0.1 increments on a 1 to 5-point scale and was immediately obtainable in the farms’ DelPro Farm Manager system (DeLaval International AB, Tumba, Sweden).

### Statistical Analysis 

Statistical analyses were performed using SAS 9.3 (SAS Institute Inc., Cary, NC, USA). All of the descriptive statistics used to stratify the factors related to ABCS were determined utilizing PROC MEANS and PROC UNIVARIATE. Body condition score data were used from 0 to 300 DIM due to the limited availability of cows to score after this threshold. Daily ABCS were matched to the corresponding DIM for each cow in the analysis throughout one lactation period per cow. Lactation number, DIM, and disease status were obtained from data integration software (Bovisync, Dairy LLC, Eden, WI, USA), which synced the data entered by farm personnel from the on-farm computer. Milk production data were gathered from DelPro Farm Manager software (DeLaval International AB, Tumba, Sweden). Outliers for the ABCS data set were identified as three SD from the median and were removed. The omitted body condition score data outliers at ABCS ≤ 2.2 and BCS ≥ 4.1 represented 0.11 and 0.13% of the total scores, respectively, yielding a data set containing 99.76% of all scores collected. Additionally, the mean daily ABCS were stratified and modeled by lactation number (1, 2, 3, 4, and ≥5), calving ABCS (≤3.2, 3.3, 3.4, 3.5, or ≥3.6), and disease status (“diseased” or “healthy”). Positive disease status was allocated if the cows developed metritis, retained placenta, or milk fever within 14 DIM, or ketosis or displaced abomasum within 30 DIM. Cows were considered diseased if they contracted one or more illnesses. All disease diagnoses were recorded in the farm logbook by farm staff or veterinarians and were obtained at the end of the study period. Lastly, the mean daily ABCS was plotted against the mean daily milk yield. Data were evaluated to determine the future ABCS to nadir. Separate univariate linear regression models were created using the PROC GLM for each evaluated explanatory variable. The variables included were DIM, lactation number, calving ABCS, calving month, diseased or healthy status, and 305PMY. Variables with *p* < 0.05 in the univariate models were offered to the multivariable model. The relationship of ABCS with all combined factors was analyzed using the MIXED procedure, with days as a repeated measureand cow as a random effect. Variables were retained in the multivariable model if *p* < 0.05, using a backwards elimination process. The following describes the function from the final multivariable mixed model:Y_ijklm_ = β_0_ + β_1_ × DIM_i_ + β_2_ × LACT_j_ + β_3_ × CABCS_k_ + β_4_ × DIS_l_ + β_5_ × PMY_m_ + *e*_ijklm_
where Y_ijklm_ is the response variable of automated body condition, β_0_ is the intercept, β_1_ is the regression days in milk, DIM_i_ is effect of days in milk, β_2_ is the regression coefficient of lactation number, LACT_j_ is the effect of lactation (lactation = 1, 2, 3, 4, and ≥5), β_3_ is the regression coefficient of calving automated body condition score, CABCS_k_ is the effect of calving automated body condition score, β_4_ is the regression coefficient of disease status, DIS_l_ is the effect of disease status (positive or negative), β_5_ is the regression coefficient of 305-d predicted milk yield, PMY_m_ is the effect of 305-d predicted milk yield, and *e*_hijklm_ is the residual error.

## 3. Results 

The distribution of all collected ABCS (*n* = 561,228) is descriptively displayed with a moving average trendline based on two stratified ABCS groups in Figure 1. The mean ABCS for all scores collected during calving and the lactation period was 3.42 ± 0.22 and 3.29 ± 0.25, respectively (range 2.2 to 4.0). The average time for cows to reach nadir was 38 days for primiparous cows and 54 days for multiparous cows while the average ABCS loss was 0.24 (±0.25) points post-calving (Table 1). The average time for cows to regain lost ABCS post-calving was 256 days (3.42 ± 0.23). The average ABCS scored at the end of the study at 300 DIM was 3.47 ± 0.22 (Figure 2a). When cows were stratified by lactation number, a similar ABCS path was seen across lactation for all lactations (Figure 2b). The mean calving ABCS was 3.13 (±0.21), 3.38 (±0.25), 3.44 (±0.29), 3.45 (±0.26), and 3.42 (±0.30 ABCS) for lactation numbers 1, 2, 3, 4, and ≥5, respectively. Cows in their first lactation had numerically less ABCS loss and consistently stayed heavier across lactation (Figure 2b). Multiparous cows lost more than twice the calving body condition percentage compared to primiparous cows by nadir, although they reached nadir 16 days later (Table 1).

Descriptively, no difference was observed in ABCS for cows that remained healthy (3.42 ± 0.21 ABCS) or developed a disease (3.41 ± 0.23 ABCS). Healthy cows reached nadir at 65 DIM at 3.18 ± 0.23 ABCS and cows that developed a disease reached nadir earlier at 59 DIM at 3.12 ± 0.25 ABCS. The average body condition loss from calving to nadir was 0.24 and 0.29 for healthy and diseased cows, respectively. By 300 DIM, healthy cows were at 3.48 ± 0.22 ABCS and diseased cows were at 3.44 ± 0.20 ABCS. When stratified by calving ABCS, post-calving nadir ABCS was reached at 46 (3.05 ± 0.25), 76 (3.10 ± 0.25), 69 (3.13 ± 0.24), 53 (3.21 ± 0.27), and 56 (3.29 ± 0.30) DIM for calving ABCS categories ≤ 3.2, 3.3, 3.4, 3.5, and ≥ 3.6, respectively (Figure 2c). 

When ABCS was plotted against milk yield, the negative energy balance associated with ABCS mobilization could be observed around 16 DIM, with a positive energy balanced reached around day 230 DIM (Figure 3). 

The mean and SD of all parameters used in individual univariate analysis are included in Table 2. Days in milk, lactation number, calving ABCS, calving month, disease status, and 305PMY were all significant predictors of ABCS in each of their individual univariate models (*p* < 0.0001; Table 3). Both DIM and calving ABCS had higher R^2^ values of 0.11 and 0.16, respectively (Table 3). When variables were used in the full multivariate model, the calving month was not significant (*p* > 0.05) and was thus removed from the model. All variables remaining in the multivariate model were significant (*p* < 0.001; Table 4).

The final multivariate model that best explained the ABCS curve throughout lactation is described as follows:ABCS_ijk_ = 1.4838 − 0.00452 × DIM_i_ − 0.03851 × Lactation number_j_ + 0.5970 × Calving ABCS_k_ + 0.02998 × Disease Status(neg)_l_ − 1.52 × 10^−6^ × 305PMY_m_ + eijkl_m_.

## 4. Discussion

The ABCS curve through lactation found in this study, utilizing an automated BCS system, followed recent modeling of BCS curves using manual scoring [40,41]. Descriptively, the mean calving ABCS for the cows in this study (3.42 ± 0.24) was higher compared to calving BCS found in a study using cows housed and fed indoors (2.92 ± 0.14) or grazed on pasture (2.87 ± 0.14) [42]. Our calving ABCS was also higher than the recommended calving BCS of 2.75 in earlier work [26,43], but lower than the calving BCS of 4.6 that was approximated in a study on New Zealand Holstein cows [44]. However, a similar mean calving BCS to our study has been reported [43]. Our results showed an average ABCS loss of 0.14 and 0.30 at 38 and 54 DIM for primiparous and multiparous cows, respectively. Our results are similar to previous findings of average losses of 0.17 BCS at 6 weeks (42 DIM) and 0.22 BCS at 8 weeks (56 DIM) post-calving in multiparous cows [45,46]. However, previous studies have reported an average body condition loss of 0.8 between calving and nadir at 50 DIM from cows from 13 Holstein herds in Prince Edward Island, Canada, and a 0.72 body condition loss between calving and nadir at 48 DIM for 897 Holstein cows in New Zealand [18,47]. It is possible that contrasting results in BCS loss between studies may be influenced by factors such as genetics, level of milk production, nutrition differences, and the addition of pasture use as a source of nutrition, which affects DMI and energy intake [48,49]. However, it is also reasonable to argue that studies based on manual body condition scoring are at risk of bias due to factors such as experience levels, training, and human error. Alternatively, ABCS is able to provide daily scores for each cow, thus increasing the precision of the condition status at any given time. It has also been argued that a quarter point change in BCS cannot be accurately evaluated by the difference in two observations alone. Instead, it is recommended that two observations are conducted for each time point, which is a task that could be easily implemented through use of ABCS systems [2,50]. The modeled and estimated ABCS curves in this study could allow for observations to be made regarding factors which potentially affect the progression of ABCS throughout lactation. Therefore, investigating these effects further could provide useful information for incorporation into predictive models for ABCS or for assisting in management on-farm.

The initial univariate models indicated that all individual parameters used were originally associated with the predictive outcome of ABCS. Calving month was not significant when entered in the full multivariate model, likely because the other parameters accounted for this variability within the calving month. It has been noted that three different periods within the lactation period have different uses and applications of BCS—early lactation for disease risk, middle lactation for breeding, and late lactation and dry period for next lactation preparation [51,52]. However, this study chose to only use the initial drop in ABCS after parturition as the predictor phase. The single phase (DIM 1 to 71) chosen for this herd incorporated both disease and the end of the voluntary waiting periods. Incorporating a predictor function into the ABCS system would allow for an accurate, individual on-farm management strategy for specific cows of interest. For example, it has been recommended to only inseminate cows with a BCS > 2.5, a threshold that could be managed on an individual cow level using an automated alert through an ABCS software to facilitate feed and reproductive management [42,53]. A possible issue with managing ABCS is the potential genetic predetermination of BCS and other body tissues, as hypothesized by previous authors [32,54,55,56]. The importance of the early lactation monitoring revolves around environmental limitations to allow the cow to reach its full genetic potential [55]. In late lactation, cows are less energy stressed and can biologically attempt to re-establish their genetic path, although they may be at risk of over-conditioning [57,58]. Genetic aspects of body condition and mobilization affect the cow’s predisposed BCS path, while the cow’s environment affects the ability to stay on that path. An early study reported that cows with a lower BCS at calving direct more DMI than body mobilization towards milk production, resulting in increased efficiency [29]. Genetic merit also affects the condition lost from calving to AI, with high merit cows losing more BCS and achieving their nadir BCS later in DIM [59,60]. The Wilmink function, described by McCarthy and others, incorporated three phases involving the curve height: initial lactation phase, final phase, and DIM [33]. As a result, their equation represented the majority of ABCS variation, which was higher than what was found in this study. This difference can be explained, in part, by the different variables used in each model. In addition, the use of a Roche−Berry−Boston (RBB) function, which has an additional late incline phase in late lactation and was included in the total function model in a study with similar objectives, may have yielded a different result compared to our models, which only investigated the initial drop in ABCS as a predictor phase for the modeling [61]. A limitation of this study was the lack of available information regarding disease duration, which would have been useful to factor into the modeling. Underlying disease is likely to inhibit BCS gain and potentially increase the rate of BCS loss [62]. 

Although this study was able to find various factors that can be used to predict future automated body condition scoring, incorporating a predictive ABCS function into farm management should still be studied. The model estimates a small portion of the factors that may influence the future body condition of cows in commercial settings and should be interpreted carefully.

## 5. Conclusions

Body condition score can vary depending on many aspects relative to a specific cow. Although other studies have evaluated and observed the impact and progression of BCS across lactation, this study aimed to determine these effects with a new, commercially available automated body condition scoring system. The constant BCS monitoring associated with ABCS may allow it to provide additional information compared to manual BCS. We identified factors which are significant for predicting the BCS curve during lactation. These could be used to monitor deviations or benchmark ABCS in lactating dairy cows. Applying ABCS into future studies on commercial dairies may assist in providing protocols regarding the management of an automated BCS system. It may also help formulate new energy requirement equations to alleviate the negative energy balance associated with the transition period. Future studies should continue using ABCS systems for BCS data collection and include additional factors, such as genetics, that may improve the accuracy of the model to facilitate applicability on-farm. Finally, ABCS may be used to help elevate the understanding of body condition changes in dairy cows. This new knowledge may be used to establish benchmarks for collecting frequent qualitative BCS data for predictive modeling and for animal welfare level assessments in certification programs.

## Figures and Tables

**Figure 1 animals-12-00601-f001:**
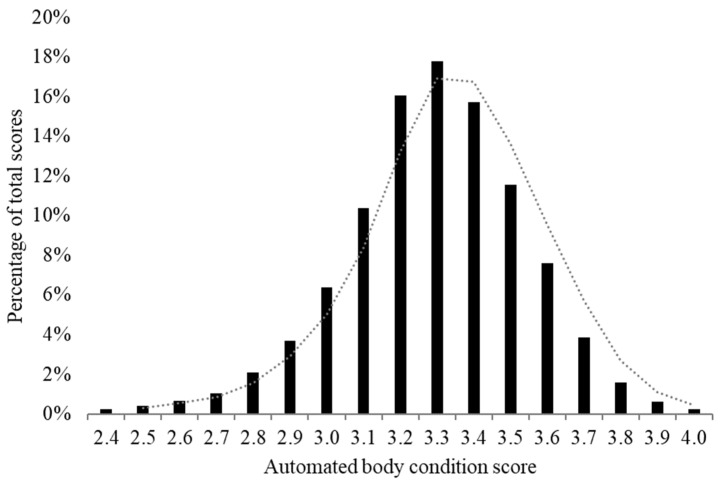
Distribution of all automated body condition scores (*n* = 561,237) collected from 2345 Holstein cows at a commercial dairy in Indiana. Dotted line shows a two-level moving average trendline across two stratified ABCS-levels.

**Figure 2 animals-12-00601-f002:**
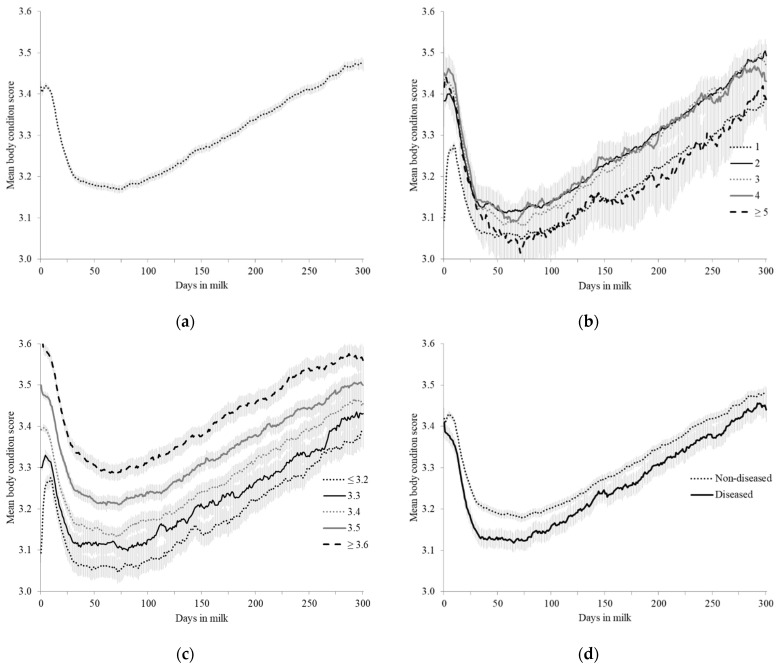
Mean (95% CI) automated body condition scores (ABCS) from 2345 Holstein dairy cattle collected using a 3D camera system at a commercial dairy in Indiana, USA. Data presented across days in milk to 300 days in milk (DIM) stratified by: (**a**) overall scores for cattle, (**b**) lactation number, (**c**) calving ABCS, and (**d**) disease status ^1^. ^1^ Disease status = positive if cows developed metritis, retained placenta, or milk fever ≤ 14 DIM, or ketosis or displaced abomasum ≤ 30 DIM.

**Figure 3 animals-12-00601-f003:**
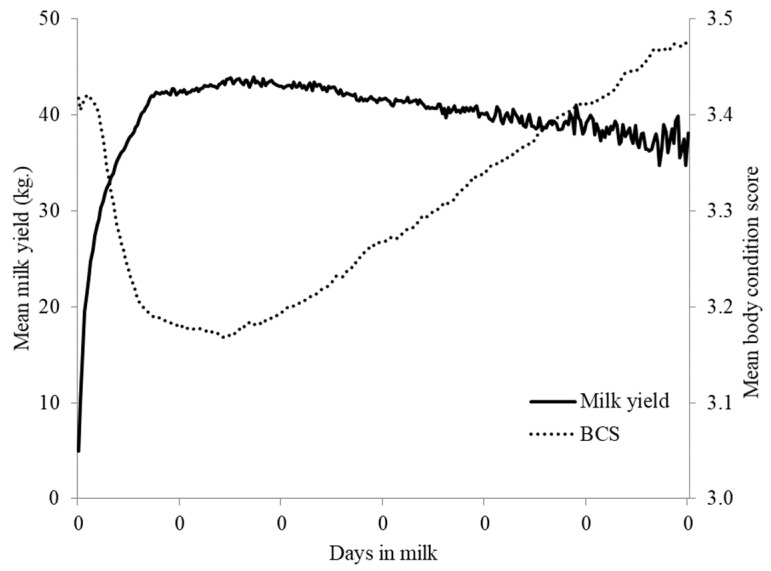
Mean automated body condition score (ABCS) and mean daily milk yield for 2345 dairy cattle collected using a 3D camera system at a commercial dairy in Indiana, USA. Data presented across days in milk to 300 days in milk (DIM).

**Table 1 animals-12-00601-t001:** Average calving automated body condition scores (ABCS), ABCS loss from calving to nadir ^1^, days to nadir, and percentage of ABCS lost to nadir for lactating dairy cows, stratified by primiparous (lactation = 1) and multiparous (lactation ≥ 2) of ABCS collected using a 3D camera system at a commercial dairy in Indiana, USA.

Parameter	Primiparious	Multiparous
Calving ABCS	3.40	3.40
Nadir ABCS	3.26	3.10
Days to nadir	38	54
ABCS loss (%)	−0.14 (−4.12)	−0.3 (−8.82)

^1^ Defined as the first lowest day by the 100th decimal place.

**Table 2 animals-12-00601-t002:** Descriptive statistics of automated body condition scores (ABCS) of lactating dairy cows collected using a 3D camera system at a commercial dairy in Indiana, USA.

Parameter	Mean	SD	Minimum	Maximum
DIM	158.10	110.2	0	505
Lactation Number	2.07	1.11	1	7
Calving ABCS ^1^	3.42	0.24	1.55	5.00
Calving Month	6.2	4.4	1	12
Disease Status ^2^	0.17 ^4^	0.37	0	1
305PMY ^3^	12,720	2028	7418	22,621

^1^ ABCS = automated body condition score; ^2^ disease status = positive if cows developed metritis, retained placenta, or milk fever ≤ 14 DIM, or ketosis or displaced abomasum ≤ DIM; ^3^ 305PMY = 305-d predicted milk yield; ^4^ 16.5% of data were from disease status positive cows.

**Table 3 animals-12-00601-t003:** The results of individual variable univariate model associations with an automated body condition score (ABCS) lactation curve of dairy cattle using a 3D camera system at a commercial dairy in Indiana, USA.

Parameter	Intercept	Estimate ^4^	SE ^5^	R ^2^	*p*-Value ^6^
DIM	3.38	−0.0038	<0.0001	0.11	<0.0001
Lactation Number	3.33	−0.043	0.00054	0.040	<0.0001
Calving ABCS ^1^	1.74	0.44	0.0026	0.16	<0.0001
Calving Month	0.16	0.00015	<0.0001	0.0064	<0.0001 *
Disease Status ^2^	3.20	0.062	0.0016	0.0095	<0.0001
305PMY ^3^	3.34	−0.327 × 10^−5^	<0.0001	0.0090	<0.0001

^1^ ABCS = automated body condition score; ^2^ disease status = positive if cows developed metritis, retained placenta, or milk fever ≤ 14 DIM, or ketosis or displaced abomasum ≤ 30 DIM; ^3^ 305PMY = 305-d predicted milk yield; ^4^ estimate for disease status refers to no negative disease status, positive disease status estimate is zero; ^5^ standard error is for parameter estimates; ^6^ significance declared for parameter estimates; * non-significant intercept.

**Table 4 animals-12-00601-t004:** Results of individual variables in a multivariate model association with an automated body condition score (ABCS) lactation curve of dairy cattle using a 3D camera system at a commercial dairy in Indiana, USA.

Parameter	Estimate ^4^	SE	*p*-Value
Intercept	1.48	0.054	<0.0001
DIM	−0.0045	<0.0001	<0.0001
Lactation Number	−0.038	0.0029	<0.0001
Calving ABCS ^1^	0.60	0.015	<0.0001
Disease Status ^2^	0.030	0.0083	0.0003
305PMY ^3^	−1.52 × 10^−6^	0.00	<0.0001

^1^ ABCS = automated body condition score; ^2^ disease status = positive if cows developed metritis, retained placenta, or milk fever ≤ 14 DIM, or ketosis or displaced abomasum ≤ 30 DIM; ^3^ 305PMY = 305-d predicted milk yield; ^4^ estimate for disease status refers to no negative disease status, positive disease status estimate is zero.

## Data Availability

The data presented in this study are available on request from the corresponding author.

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
