# Peer review of "Body Condition Score Change throughout Lactation Utilizing an Automated BCS System: A Descriptive Study"

_animals, 2022, doi:10.3390/ani12050601_

Round 1
Reviewer 1 Report
In this paper, the Authors determine the different trajectories of BCS derived from ABCS cameras by investigating a multivariate modelling approach including variables (DIM, lactation number, calving ABCS, calving month, disease, and 305-d predicted milk yield)
The title indicates the aim of the manuscript and the abstract is well written. The introduction is well written. The objectives of the study are of interest and are perfectly in line with the scope of the journal.
The manuscript is well organized. The methodology is well articulated and the description is well made.
The discussion and conclusions are consistent with the evidence and arguments presented and they address the main question posed.
The reference is appropriate.
I have one doubt: Were informed farmers/owners about study objectives and procedures? A written informed consent prior to enrolment should be obtained from farmers/owners. According to journal's guideline, for studies involving client-owned animals written informed consent must be obtained from the farmer/owner of the animals (or an authorized agent for the owner).
I recommend its publication after the aforementioned revison.
Author Response
Reviewer comments:
Reviewer 1:
In this paper, the Authors determine the different trajectories of BCS derived from ABCS cameras by investigating a multivariate modelling approach including variables (DIM, lactation number, calving ABCS, calving month, disease, and 305-d predicted milk yield)
The title indicates the aim of the manuscript and the abstract is well written. The introduction is well written. The objectives of the study are of interest and are perfectly in line with the scope of the journal.
The manuscript is well organized. The methodology is well articulated and the description is well made.
The discussion and conclusions are consistent with the evidence and arguments presented and they address the main question posed.
The reference is appropriate.
I have one doubt: Were informed farmers/owners about study objectives and procedures? A written informed consent prior to enrolment should be obtained from farmers/owners. According to journal's guideline, for studies involving client-owned animals written informed consent must be obtained from the farmer/owner of the animals (or an authorized agent for the owner).
I recommend its publication after the aforementioned revision.
AU: We thank the reviewer for the kind feedback of the manuscript. Yes, the farmer was aware about the study and we have provided a written and signed consent form with the revised manuscript.
Reviewer 2:
I thought this was a good paper describing how new technology can improve farming systems and I fully support the authors conclusions.
AU: We appreciate your thoughtful comments, time taken to revise the manuscript, and positive view of this work. We have updated the manuscript in regards to your and other reviewer comments to the best of our ability.
Specific comments;
Materials and methods;
The automated system must include a method of cow identification.
AU: This is correct, information has been added on lines 128-131 in the revised manuscript.
The cow feeding system should ay least be mentioned. Is it a barn fed, total mixed ration fed to all cows and is there any adjustment of rations due to such things as stage of lactation, time of year or milk yield etc?
AU: This information has been added on lines 123-125.
the ABCS system seems to calibrate on the expected points on the cow. The Edmonson et al (1989) reference is given, and does give a description of what the system is scoring to allow people who are not used to ABCS an idea of what the cows are like.
Confirm that each set of records was only 1 lactation per cow (i.e. no repeat lactations)
AU: This information has been added.
The calving ABCS : was that simply the first day the cow came through the milking shed or was there some pre calving assessment, if so how? I was also a bit intrigued to see the data that ABCS increased for a few days after calving.
AU: Yes, there was the first day of lactation.
Under disease status lameness does not seen to be a factor - probably a reflection of the herd.
Results: As we were using farm diagnose, the events of lameness and mastitis were included in the other events. Different from the transition disease that were identified by the research group.
I am not sure what is meant by "a moving average trendline based on 2 stratified ABCS groups in Fig. 1"
AU: This was performed to show to the reader the trend trough time, and to smother any major change in the BCS trough time.
Where is Fig 3?
AU: Figure 3 was somehow missed to be added during the submission. We apologize and we have added the figure to the revised manuscript.
Discussion: I thought the comment that the references to grazing measurements for loss of BCS between calving and the nadir are much larger than seen here - my comment is that this type of ABCS might be even more useful in pasture based systems.
AU: Thank you for this comment. Indeed, this automated system has an outdoor version capable of being used in pasture based systems. More information can be found on DeLavals homepage as well in a few YouTube clips online if you search for DeLaval automated body condition scoring. Well worth a view.
Reviewer 3:
The aim of the study was to implement a commercially available automated body condition scoring (ABCS) camera system to collect data for developing a predictive equation of body condition dynamics throughout the lactation period. The introduction is well written but is too long and inapropriate. The objective of that work is not the study of BCS during the lactation period but the use automated body condition scoring. I suggest to improve the section of past studies about this topic.
AU: We appreciate the comments on the introduction. We do agree to some extent that parts of the introduction may be somewhat redundant in describing earlier work on BCS in general, in comparison to the question investigated. However, we also feel that it is important to explain previous BCS studies and the current ambiguity that lies herein as a rationale why automation and increased precision and accuracy is of great importance. As neither of the other reviewers nor editor has commented on the current introduction content nor length, we have opted to keep it as is for the time being. We hope this is ok with the reviewer.
The results are well discussed and clearly presented. The discussions are complete and explain in a good way the results obtained
AU: Thank you very much.
Reviewer 4:
This an interesting and well-prepared study as well as a good manuscript.
It is a pity that some biases and some confounders are present in this study. Nevertheless, it is worth to be published in ANIMALS.
The whole study was performed on one site, so the between-effect of farms cannot be addressed but should be discussed.
AU: We appreciate your time spent reviewing this manuscript and your comments that will help improve the manuscript. We agree that it is unfortunate that there are some biases and that only one farm was enrolled. However, as it was a collaboration with a private company, we had to adhere to certain restrictions imposed on the project, and this was to start with one large herd on one site. We do appreciate the sentiment that there is value in publishing this data, and we obviously agree on this point.
Specific comments:
Line 29: Reduce the formula without the parentheses.
AU: We have removed the parentheses in the formula in the abstract and in the results section.
Line 41: Explain intra and inter variation.
AU: We added a brief explanation in the introduction.
Line 86: Here You have to mention the delay of the actual BCS and the ongoing of a problem such as CRA (chronic rumen acidosis)
AU: We have now mentioned this is the introduction.
Line 108: When You have an ABCS by camera which gives You only a delayed status of the cow under investigation, why do You not measure the RFS (rumen filling score) to get an actual value of the cow and connect this 2 informations?
AU: We did not have the possibility to measure daily RFS on 2,343 cows with the current study setup. This was a commercial farm, where the ABCS system was left to run for the duration of the trial. However, all cows were scored before the first feeding session of the day.
Line 115: Kg >> kg.
AU: This has been corrected.
Line 147: What’s about mastitis or lameness?
AU: As we were using farm diagnose, the events of lameness and mastitis were included in the other events. Different from the transition disease that were identified by the research group.
Line 152: Didn’t You not collect data on duration of disease? This may influence BC more than different diseases.
AU: We were unable to collect any data on disease duration. This study was a retro-active study using already collected data and farm records regarding disease length for individual cows were inaccessible to us. We had to resort to report the number of reported diseases.
Line 155: Which is the repeated measure?
AU: The repeated measure was day. It was stated on line 156 in the old manuscript.
Line 220: Eliminate the parentheses.
AU: This has been corrected.
Line 223: Check column size in SD.
AU: This has been formatted to a better width.
Line 280: As mentioned above, the duration of disease can explain more of the variation of ABCS than only disease. Since I guess You don’t have the data of duration of disease, You have to discuss it anyway.
AU: We have added this as a limitation to the current study at the end of the discussion.
Line 309: Maybe You have to discuss the economic of ABCS. What is the actual prize of ABCS? What’s about a rumen online device connected with ABCS? The merged data could provide more insights into the health of a cow.
AU: As our objectives were to create a standardized ABCS curve and identify the factors that are influential, we refrain to look at the economics and how the camera can be used.
Reviewer 5.
This study used automated imaging technology for cow body condition scoring to establish a commercial Automated Body Condition Scoring (ABCS) camera system. The reported results are interesting and possibly novel in the field, however, the Materials and Methods and Results section of the manuscript is incomplete. I have the following specific comments about the submitted manuscript.
AU: Thank you very much for the insightful comments and suggestions, we have tried to work them into the manuscript alongside other reviewer comments to the best of our ability.
Specific comments:
Line 117-118: No details on how to ensure that all lactating cows are put through the camera alone once a day? Is each cow identified using an electronic ear tag or by other means? Imaging was acquired only once a day. Whether images were taken before or after feeding, was not specified.
AU: Thank you for pointing this out. All cows were identified using RFID tags and cows were put through the camera post-morning milking but before the first feeding of the day. This information has been added to the manuscript.
Line 119-125: If cows have lameness or other hoof diseases, whether passing the camera will affect the accuracy of photo scoring.
AU: It is possible that the BCS scoring may have been affected by any locomotion differences if it was based on one image. However, as a video was taken and each key characteristic was measured multiple times by each image taken within the video, any discrepancies would have been mitigated by the multiple measurements.
Line 146-148: Please clarify how positive diseases were diagnosed. With veterinary detection, or with 3D cameras.
AU: All diseases were recorded by staff or farm veterinarian in the farm log book. We obtained disease records at the end of the study. This information has been added.
Line 159: The relationship between the equation in L159 and the related equation in the abstract is not explained in detail, and the content is not clear. In addition, are the functional equations involved in the manuscript only applicable to the ABCS data obtained by 3D photography, and are they also apply to the data processing obtained by still photos, thermal imaging, and optical depth cameras?
AU: This study is focused on the technology used and we can not make inferences about the other technologies. We have included in the discussion a sentence suggesting future research on these topics.
Line 171-175: The result description does not correspond to the data involved in Table 1, and there are errors in the result description or data. It is not explicitly shown in Table 1 that the average ABCS loss was 0.24 (± 0.25) points post-calving.
AU: Corrected.
Line 186-187: Please clarify how are the two ABCS levels determined?
AU: The second stratification is a moving average of the ABCS collected.
Line 207: Figure 2d is not described in the experimental results.
AU: Thank you for bringing this to our attention. This must have been accidentally omitted during a previous round of edits. We have added this information.
Line 209: Figure 3?
AU: Figure 3 ended up not getting submitted. We apologize for this and we have added figure 3 to the revised manuscript.

Reviewer 2 Report
I thought this was a good paper describing how new technology can improve farming systems and I fully support the authors conclusions.
Specific comments;
Materials and methods;
The automated system must include a method of cow identification.
The cow feeding system should ay least be mentioned. Is it a barn fed, total mixed ration fed to all cows and is there any adjustment of rations due to such things as stage of lactation, time of year or milk yield etc?
the ABCS system seems to calibrate on the expected points on the cow. The Edmonson et al (1989) reference is given, and does give a description of what the system is scoring to allow people who are not used to ABCS an idea of what the cows are like.
Confirm that each set of records was only 1 lactation per cow (i.e. no repeat lactations)
The calving ABCS : was that simply the first day the cow came through the milking shed or was there some pre calving assessment, if so how? I was also a bit intrigued to see the data that ABCS increased for a few days after calving.
Under disease status lameness does not seen to be a factor - probably a reflection of the herd.
Results;
I am not sure what is meant by "a moving average trendline based on 2 stratified ABCS groups in Fig. 1"
Where is Fig 3?
Discussion: I thought the comment that the references to grazing measurements for loss of BCS between calving and the nadir are much larger than seen here - my comment is that this type of ABCS might be even more useful in pasture based systems.
Author Response

(The authors gave the same response as above.)

Reviewer 3 Report
The aim of the study was to implement a commercially available automated body condition scoring (ABCS) camera system to collect data for developing a predictive equation of body condition dynamics throughout the lactation period. The introduction is well written but is too long and inapropriate. The objective of that work is not the study of BCS during the lactation period but the use automated body condition scoring. I suggest to improve the section of past studies about this topic.
The results are well discussed and clearly presented. The discussions are complete and explain in a good way the results obtained
Author Response

(The authors gave the same response as above.)

Reviewer 4 Report
General:
This an interesting and well-prepared study as well as a good manuscript.
It is a pity that some biases and some confounders are present in this study. Nevertheless, it is worth to be published in ANIMALS.
The whole study was performed on one site, so the between-effect of farms cannot be addressed but should be discussed.
Comments
Line |
Comment |
29 |
Reduce the formula without parentheses. |
41 |
Explain intra and inter variation. |
86 |
Here You have to mention the delay of the actual BCS and the ongoing of a problem such as CRA (chronic rumen acidosis) |
108 |
When You have an ABCS by camera which gives You only a delayed status of the cow under investigation, why do You not measure the RFS (rumen filling score) to get an actual value of the cow and connect this 2 informations? |
115 |
Kg >> kg |
147 |
What’s about mastitis or lameness? |
152 |
Didn’t You not collect data on duration of disease? This may influence BC more than different diseases. |
155 |
Which is the repeated measure? |
220 |
eliminate paranthesis |
223 |
check column size in SD |
280 |
As mentioned above, the duration of disease can explain more of the variation of ABCS than only disease. Since I guess You don’t have the data of duration of disease, You have to discuss it anyway. |
309 |
Maybe You have to discuss the economic of ABCS. What is the actual prize of ABCS? |
309 |
What’s about a rumen online device connected with ABCS? The merged data could provide more insights into the health of a cow. |
Author Response

(The authors gave the same response as above.)

Reviewer 5 Report
This study used automated imaging technology for cow body condition scoring to establish a commercial Automated Body Condition Scoring (ABCS) camera system. The reported results are interesting and possibly novel in the field, however, the Materials and Methods and Results section of the manuscript is incomplete. I have the following specific comments about the submitted manuscript.
Specific comments:
Line 117-118: No details on how to ensure that all lactating cows are put through the camera alone once a day? Is each cow identified using an electronic ear tag or by other means? Imaging was acquired only once a day. Whether images were taken before or after feeding, was not specified.
Line 119-125: If cows have lameness or other hoof diseases, whether passing the camera will affect the accuracy of photo scoring.
Line 146-148: Please clarify how positive diseases were diagnosed. With veterinary detection, or with 3D cameras.
Line 159: The relationship between the equation in L159 and the related equation in the abstract is not explained in detail, and the content is not clear. In addition, are the functional equations involved in the manuscript only applicable to the ABCS data obtained by 3D photography, and are they also apply to the data processing obtained by still photos, thermal imaging, and optical depth cameras?
Line 171-175: The result description does not correspond to the data involved in Table 1, and there are errors in the result description or data. It is not explicitly shown in Table 1 that the average ABCS loss was 0.24 (± 0.25) points post-calving.
Line 186-187: Please clarify how are the two ABCS levels determined?
Line 207: Figure 2d is not described in the experimental results.
Line 209: Figure 3?
Author Response

(The authors gave the same response as above.)
